# Unraveling the Molecular and Cellular Pathogenesis of COVID-19-Associated Liver Injury

**DOI:** 10.3390/v15061287

**Published:** 2023-05-30

**Authors:** Hikmet Akkiz

**Affiliations:** Department of Gastroenterology and Hepatology, Medical Faculty, Bahçeşehir University, Istanbul 34349, Turkey; hakkiz@superonline.com

**Keywords:** SARS-CoV-2, COVID-19, liver injury, chronic liver disease, autoimmune liver disease, vaccination

## Abstract

The coronavirus disease 2019 (COVID-19) pandemic, caused by severe acute respiratory syndrome coronavirus (SARS-CoV-2) continues to cause substantial morbidity and mortality. Most infections are mild; however, some patients experience severe and potentially fatal systemic inflammation, tissue damage, cytokine storm, and acute respiratory distress syndrome. Patients with chronic liver disease have been frequently affected, experiencing high morbidity and mortality. In addition, elevated liver enzymes may be a risk factor for disease progression, even in the absence of underlying liver disease. While the respiratory tract is a primary target of SARS-CoV-2, it has become evident that COVID-19 is a multisystemic infectious disease. The hepatobiliary system might be influenced during COVID-19 infection, ranging from a mild elevation of aminotransferases to the development of autoimmune hepatitis and secondary sclerosing cholangitis. Furthermore, the virus can promote existing chronic liver diseases to liver failure and activate the autoimmune liver disease. Whether the direct cytopathic effects of the virus, host reaction, hypoxia, drugs, vaccination, or all these risk factors cause liver injury has not been clarified to a large extent in COVID-19. This review article discussed the molecular and cellular mechanisms involved in the pathogenesis of SARS-CoV-2 virus-associated liver injury and highlighted the emerging role of liver sinusoidal epithelial cells (LSECs) in virus-related liver damage.

## 1. Introduction

In December 2019, severe acute respiratory coronavirus syndrome 2 (SARS-CoV-2) was initially identified as a novel pathogen in Wuhan, China, that rapidly spread throughout the world, leading to the COVID-19 pandemic [1,2]. SARS-CoV-2 causes upper and lower respiratory tract infections that are frequently associated with fever, cough, and loss of smell and taste [1,2]. Most patients have mild disease, and approximately 20–40% of patients are asymptomatic [1,2,3]. However, some patients experience more severe disease and develop systemic inflammation, tissue damage, acute respiratory stress syndrome, thromboembolic complications, cardiac injury, and/or cytokine storm, which can be fatal [3,4]. The risk of COVID-19 disease severity depends on comorbidities, including diabetes, hypertension, and obesity, along with age, ethnicity, genetic factors, vaccination status, and other conditions [2,3,4,5]. The disease consists of an early infection phase in which the virus enters pulmonary epithelial cells through the surface ACE2 receptor, causing viral pneumonia, followed by a systemic inflammatory phase characterized by respiratory failure and multiorgan dysfunction [6,7].

SARS-CoV-2 is an enveloped, positive-sense, single-stranded RNA virus that is a member of the Betacoronavirus genus in the Coronaviridae family, and it is related closely to SARS-CoV and Middle East respiratory syndrome CoV [3,8,9,10]. The SARS-CoV-2 virus encompasses a large genome of about 30 kbp coding for 16 nonstructural and 4 structural proteins including spike (S), envelope (E), membrane (M), and nucleocapsid (N) [8,11,12]. The spike protein is a type I glycoprotein that forms peplomers on the virion surface, and it plays a key role in the life cycle of the SARS-CoV-2 virus, including viral attachment, fusion, entry, and transmission [11,12,13,14]. The spike protein has two functional components: the S1 and S2 domains [8,11,12,13]. The S1 domain mediates receptor binding and the S2 mediates downstream membrane fusion. The receptor binding domain (RBD) located in the S1 unit is the most variable part of the coronavirus genome [11,12]. For infection of most host cells, the SARS-CoV-2 S protein binds to its main cellular receptor, angiotensin-converting enzyme 2 (ACE2) [8,11,12,13]. Additionally, the host transmembrane proteases serine 2 (TMPRSS2) is important for proteolytic priming of the S protein for receptor interaction and entry. Other host proteins, such as neuropilin-1, heparin sulfate proteoglycans, C-type lectins, or furin, can also act as co-factors for viral entry [3,11,12]. Spike protein binds to a specific host cellular receptor (ACE2) and host proteases such as TMPRSS2 that promote viral uptake and fusion [11,12]. ACE2 and TMPRSS2 are aberrantly expressed in airways, lungs, nasal/oral mucosa, and the intestine [7,10,11]. The binding affinity of the spike protein to the ACE2 receptor affects the SARS-CoV-2 replication fitness and disease severity [11,12,13,15,16]. Although the RBD is immunodominant, the other spike regions, particularly the NTD, play significant roles in antigenicity [11,12]. Researchers have identified four deleted regions (RDRs) within the NTD, modulating NTD antigenicity. RNA viruses have extremely high mutation rates because enzymes of the virus’s copying RNA generally lack proofreading activity [11,12,13]. SARS-CoV-2 is continuing to evolve, producing novel variants with spike protein mutations. By the end of 2021, the B.1.1.529 (omicron) SARS-CoV-2 variant displaced the B.1.617.2 (delta) variant as the predominant circulating strain in many countries [13]. Compared to earlier variants, omicron is more transmissible and resistant to neutralization by antibodies induced by current vaccine platforms or following SARS-CoV-2 infection [13,14,15,16]. See Figure 1.

Although the SARS-CoV-2 virus primarily causes significant respiratory pathology, it may lead to several extrapulmonary manifestations [17,18]. These conditions include vascular complications, myocardial dysfunction, gastrointestinal symptoms, hepatocellular and hepatobiliary injury, acute kidney injury, neurological complications, and dermatologic conditions [17,18]. Additionally, novel studies have indicated that SARS-CoV-2 could also have serious adverse effects on the reproductive system by altering sperm parameters in males and increasing the rate of pregnancy-related disorders in females, such as preeclampsia [19]. The synergistic effects of several biological mechanisms on the testis can cause damage to the testicular tissue and adversely affect male fertility, including the presence of SARS-CoV-2 in the germ cells, the effect of the virus on productive hormones, and the inflammatory response [19]. Patients with COVID-19 often have liver injury associated with adverse outcomes such as intensive care unit admission and death [20,21]. Histopathological studies of COVID-19 livers revealed an increased prevalence of moderate steatosis, mild lobular and portal inflammation, and sinusoidal thrombosis [21,22,23,24]. Although the mechanisms underlying liver injury have not been completely elucidated, direct cytotoxicity, vascular alterations, COVID-19-related immunological and inflammatory processes, COVID-19 vaccine-mediated immune responses, and drug-induced liver injury (DILI) has been implicated in the pathogenesis of COVID-19-associated liver injury [25,26,27].

The immune system is a key driver in the development, progression, and clinical outcome of COVID-19 infection [9,12,13,14,15]. Innate immune responses mitigate viral entry, translation, replication, and assembly, and they facilitate recognition and elimination of infected cells, as well as accelerate the activation of adaptive immunity [3,9,10,11,16]. Cell surface, endosomal, and cytosolic pattern recognition receptors (PRRs) respond to pathogen-associated molecular patterns (PAMPs) to stimulate inflammatory responses and programmed cell death, which mitigate viral replication and induce eradication [3,15]. On the other hand, aberrant activation of the immune system can result in systemic inflammation and severe disease [3,9,28,29]. In response to innate immune-dependent viral eradication mechanisms, coronaviruses have developed escape strategies that limit host control and foster replication [28,29,30,31]. The adaptive immune system is a main determinant of the clinical outcome after SARS-CoV-2 infection and plays an important role in the effectiveness of the vaccine [9,29,30]. T cell immunity develops early during COVID-19 infection and correlates with protection [3,9,28,29]. The T cell immune response against SARS-CoV-2 is impaired in severe disease and is associated with aberrant activation of T cell immunity and lymphopenia [10,30,32]. T cell memory comprises broad recognition of viral proteins, which recognizes about 30 epitopes within each individual [9,28,32]. This recognition capacity of T cells can mitigate the effect of viral mutations and can protect against severe disease caused by SARS-CoV-2 variants of concern, including Omicron [3,9,13,29,30]. Recent studies have indicated that the level of baseline viral load and the efficacy of the innate immune response, particularly that mediated by type 1 interferon, appear to be critical in the context of subsequent adaptive response and the clinical outcome [9,10,31,32,33,34,35].

## 2. Mechanisms Involved in Liver Injury in COVID-19

SARS-CoV-2 can lead to hepatocellular and hepatobiliary injury in the liver and many diseases as a result. All of these are considered collateral damage to the liver caused by COVID-19 [17,18,20,22,23,24]. Although there is no standard definition, liver injury is defined by abnormal liver biochemistry parameters, particularly elevations in ALT and AST levels. Among patients who are hospitalized with symptomatic COVID-19 infection, abnormal liver function tests (LFTs) are common, ranging from 3% to 53% [21,22,23,26,27]. The large variation in the frequency of liver injury in cohorts with COVID-19 reported by research groups may be due to differences in the parameter used in the definition of liver injury, threshold values, and liver injury pattern [20,21,22,23,24]. As such, there is currently an urgent need for an international definition of liver injury. Hepatic dysfunction is significantly higher in critically ill patients in the intensive care unit (ICU) who require mechanical ventilation, reaching up to 45% of cases [25,26,27,36,37]. Furthermore, some studies have reported that COVID-19 patients with elevated liver enzymes are often male, elderly, and obese [37,38]. Liver enzyme alterations can be detected in patients with or without preexisting liver disease in COVID-19 [21,25,26,27]. The most commonly detected abnormalities are hypoalbuminemia, elevated gamma-glutamyl transferase (GGT), mild elevation of aminotransferases, and hyperbilirubinemia [25,26]. Increases are usually mild (<5 times the upper limit of normal) and occur in the early stages of infection [21,26,27]. The pattern of liver injury is typically hepatocellular rather than cholestatic [36,37,38,39,40]. However, in the later stages of the disease, a progressive elevation of cholestasis parameters (alkaline phosphatase, gamma-glutamyl transferase) has been reported [21,25,26,27,36,37,38]. Liver injury generally does not result in liver failure in patients without preexisting liver disease [27,36,39,40]. However, patients with decompensated cirrhosis are at substantial risk of developing acute-on-chronic liver failure [25,26].

### 2.1. Direct Cytotoxicity

Until recently, liver injury in COVID-19 was considered to be mediated by systemic inflammation rather than the direct cytopathic effect of the virus on liver cells. Although viral proteins and RNA can be detected in the liver tissue of patients with COVID-19, active infection of parenchymal liver cells has not yet been indicated [25]. In addition, the low prevalence of clinically significant liver injury in patients without chronic liver disease (CLD) indicates that direct infection of parenchymal liver cells is unlikely to be a major mechanism of liver injury [25,26]. On the other hand, in vitro, data have demonstrated that SARS-CoV-2 can infect the hepatobiliary system and potentially can cause direct viral damage [39,41,42,43]. ACE2 and TMPRSS2 receptors, which are required for SARS-CoV-2 cellular entry, are expressed at only low levels on hepatocytes and non-parenchymal cells [13,25,39]. In a healthy liver, the highest expression of the ACE2 receptor is in the human intrahepatic biliary epithelial cells (BECs) [26,39,44,45]. Studies from liver-derived and induced pluripotent stem cell (iPSC)-derived organoids indicate that SARS-CoV-2 can infect and replicate in BECs [26,39,43,45,46,47]. Furthermore, cholangiocytes are highly susceptible to SARS-CoV-2 entry and replication [26,39,45,46]. While histopathological studies have reported the detection of SARS-CoV-2 RNA and/or proteins in human liver tissue and bile samples, it remains elusive whether SARS-CoV-2 replicates in BECs in vivo [32,40]. It can be assumed that the SARS-CoV-2 virus has a cytopathic effect on liver cells, but the route of the virus to the liver is not known [32,40]. SARS-CoV-2 may have reached the liver through the blood because, in patients with severe disease, viral RNA and virion can be detected [39,48]. In addition, the virus can reach the liver via the ascending route from the biliary tract or the portal vein as a result of translocation from enterocytes [39,48,49].

In contrast, hepatocytes express low levels of ACE2, indicating a low potential for the SARS-CoV-2 virus, although the virus can complete its life cycle in hepatoma cell lines (e.g., Huh7, HepG2) and liver-derived organoids [26,50,51]. Electron microscopic trials indicate the presence of intracellular virus particles within the hepatocyte, associated with mitochondrial swelling and structural damage, strongly suggesting a direct cytopathic effect of SARS-CoV-2 in hepatocytes [26,52,53]. A cirrhotic liver is associated with increased expression of hepatocyte ACE2; as such, the entry potential of the SARS-CoV-2 may be increased in this context. Increased hepatic expression of hepatocyte ACE2 has been indicated in patients with NAFLD and HBV-related cirrhosis [54,55]. Increased expression of ACE2 is thought to be an injury response to liver fibrosis; therefore, ACE2 was suggested to be a therapeutic target for CLD [56]. The potential mechanism for exaggerated liver injury due to SARS-CoV-2 in CLD is the increased infection burden and consequent extensive hepatocyte death. Indeed, liver-derived organoids from patients with NASH-related cirrhosis indicate remarkably increased permissiveness to SARS-CoV-2 and pro-inflammatory gene expression compared to liver organoids from non-cirrhotic donors [57]. To understand whether the liver injury in COVID-19 is due to the direct cytopathic effect of the virus or is the result of severe illness/systemic inflammation, it is useful to compare it with other severe respiratory viruses. For example, the influenza virus is considered to only infect respiratory epithelial cells, and seasonal influenza does not lead to liver injury. However, the more severe influenza A/H1N1 virus is associated with liver injury, which is correlated with a degree of hypoxia and systemic inflammation [26,58]. Recently, in another retrospective study, researchers compared influenza and COVID-19 in hospitalized patients and found similar predictors of disease severity on multivariate analysis in both groups, including aminotransferases, age, sex, and degree of systemic inflammation. As such, these data support that hypoxia or inflammation/immune-mediated mechanisms are the main drivers of liver injury in these respiratory infections [26,59].

### 2.2. Vascular Alterations following COVID-19 in Liver

One of the major concerning properties of SARS-CoV-2 infection is a coagulopathy characterized by high D-dimer and fibrinogen levels with minor changes in prothrombin time and platelet count [59,60]. This SARS-CoV-2-associated coagulopathy leads to a prothrombotic state which is highly prevalent in COVID-19, and thrombotic complications are a key cause of morbidity and mortality [61]. Detection of ACE2 receptors on vascular endothelial cells, as well as the presence of endothelitis and viral protein in endothelial cells, has increased suspicion for endothelial cell injury or activation as a central property of the pathophysiology of COVID-19, especially during the inflammatory phase of the infection [60,61,62]. On liver histology, steatosis and mild inflammatory infiltration in the hepatic lobule and portal tract have been observed in patients with COVID-19, and several studies have reported sinusoidal thrombosis [25,61,62,63,64]. Liver injury in COVID-19 has been associated with some hypercoagulable parameters, with evidence of microthrombi on liver histopathology. This finding suggests a role for vascular pathology in liver injury in COVID-19, but mechanistic details are lacking [61]. The most prevalent histopathological findings of COVID-19 livers are moderate macrovesicular steatosis and mild lobular and portal inflammation, and sinusoidal thrombosis [65,66,67]. In addition, COVID-19 has been reported to be more severe in patients with obesity and metabolic syndrome [65]. While steatosis and lobular and portal inflammation are frequent properties of metabolic liver disease, sinusoidal thrombosis may be a potential candidate for a specific feature of COVID-19-related liver injury [65]. Patients with COVID-19 present coagulopathy and endotheliopathy, characterized by elevated levels of von Willebrand factor (vWF) and soluble thrombomodulin, which has been associated with disease severity and mortality [61,66,67]. McConnel et al. has also indicated remarkable elevated activity of factor VIII, which is expressed primarily by liver sinusoidal endothelial cells (LSECs), in critically ill patients with COVID-19, suggesting a role for hypercoagulable LSECs in COVID-19-related liver injury [61,65]. Indeed, COVID-19 is a prothrombotic disease associated with a high risk of venous thrombosis, pulmonary embolism, and endotheliopathy, and sinusoidal thrombosis was also detected in liver histopathological studies of severe COVID-19 [65,68]. Studies have demonstrated that endotheliopathy and platelet activation are important features of COVID-19 in hospitalized patients [65]. Although initial studies have investigated vWB elevations exclusively in the ICU setting, Goshua et al. reported that vWB was also elevated in non-critically ill patients with COVID-19 [66]. The same study demonstrated that critical illness was associated with further elevations in vWB when compared with controls [66].

### 2.3. The Impact of Immunological and Inflammatory Processes of COVID-19 on the Liver

The robust systemic proinflammatory response is a hallmark of COVID-19, which can lead to uncontrolled massive cytokine release that can cause multi-organ failure [65,66]. Interleukin 6 (IL-6) is a pivotal proinflammatory cytokine in COVID-19 pathogenesis, and IL-6 inhibition can improve clinical outcomes and survival of patients with severe disease [65,69]. IL-6 is expressed by macrophages, endothelial cells, T cells, and fibroblasts upon stimulation of Toll-like receptor 4, IL-1, or TNF-α [61,65,66,69,70]. IL-6 stimulates downstream signaling through Janus kinase (JAK)/signal transducer and activator of transcription (STAT) activation by 2 pathways [61,65,70]. Classical IL-6 signaling is a through IL-6 binding to the ligand-binding alpha subunit of its receptor (gp80/IL-6Ra) and subsequently recruiting the signaling beta subunit (glycoprotein 130 (gp130) to induce downstream signaling [61,66,67]. This classical pathway of signal transduction restricts IL-6 signaling to cells expressing IL-6R in the liver, such as hepatocytes and cholangiocytes [61]. Trans-signaling occurs with IL-6 binding to a soluble form of the receptor alpha subunit (sIL-6R) to build an IL-6/sIL-6R complex that crosstalks with the beta subunit (gp130) on target cells, which may not produce IL-6R [61,65]. IL-6 trans-signaling is considered the main pathway of IL-6 signaling to LSECs and was implicated in endotheliopathy in COVID-19. IL-6R can be also cleaved into a soluble receptor (sIL-6R), which can build a complex with IL-6 and bind to gp130 on the cell surface of IL-6R-negative cells [65]. Baseline levels of sIL-6R are relatively high and were indicated to increase in COVID-19 [69,70,71,72,73]. Thus, IL-6 is an appealing potential mediator of endotheliopathy in the liver. Once initiated, the classical and trans-signaling of IL-6 both lead to the activation of the tyrosine kinase JAK1, MAP kinase, and STAT1 and STAT3 pathways [61,65]. Initial trials have indicated that IL-6 promotes both the acute phase response and liver regeneration, carcinogenesis, and modulation of glucose metabolism [70].

Recently, McConnell, Kawaguchi, et al. published a landmark paper adding another piece to the puzzle in our understanding of the pathogenesis of liver injury in COVID-19 [61]. Researchers first demonstrated that the most common liver pathological features in COVID-19 patients were liver congestion (98%), steatosis (47%), sinusoidal erythrocytes recruitment (44%), and neutrophil infiltration [61]. Among COVID-19 patients, patients with higher alanine aminotransferase (ALT > 3×) had higher plasma levels of IL-6 and pro-coagulation factors and the liver histopathological examination demonstrated substantially higher intralobular neutrophil infiltration and trends toward a higher prevalence of steatosis and sinusoidal erythrocyte recruitment [61]. One of the key findings of this study was that immunostaining analysis indicated that LSECs in COVID-19 patients were highly positive for vWF and demonstrated platelet recruitment at their surface [61]. Additionally, vWF production and platelet recruitment in LSECs are higher in patients with ALT < 3× and correlated with intralobular neutrophil infiltration and plasma IL-6 levels [61]. Functional trials in LSECs indicated that IL-6 trans-signaling increases the production of procoagulant factors (Factor VIII and vWF), pro-inflammatory molecules (IL-6, CXCL1, and CXCL2), and cell adhesion molecules such as ICAM1, P- and E-selectin that promote platelet attachment and neutrophil recruitment [73]. Furthermore, this trial enables evidence that activated LSECs foster the systemic inflammatory response through interacting with hepatocytes and increasing their expression of acute phase reactants such as fibrinogen [73]. IL-6 plays an important role in these processes and could be pharmacologically improved through JAK1 inhibitors that are under evaluation for the treatment of severe COVID-19. Based on their findings, LSECs respond to IL-6 through differentiating into a procoagulant and proinflammatory phenotype that stimulates platelet recruitment in sinusoids and liver neutrophil aggregation, promoting COVID-19-related liver injury [74]. Although LSECs do not produce IL-6R, the researchers reported that LSECs respond to IL-6R trans-signaling and interact with hepatocytes to express pro-coagulant and pro-inflammatory molecules [61,65]. See Figure 2.

As previously highlighted, the immune system and cytokines play a key role in the development and clinical outcome of COVID-19. Sustained high levels of some cytokines such as CXCL10, CCL7, and IL-1RA are associated with liver injury as well as fatal outcomes [16,75,76]. Effective viral control is associated with a type 1 CD4^+^ phenotype; a type 2 profile is often observed in those with severe disease [9,76,77,78]. High expression levels of effector molecules by CD8^+^ T cells in acute COVID-19 are associated with improved clinical outcomes [9,79]. However, excessively high levels of T cell activation are associated with poor clinical outcomes [80]. Furthermore, the expression of potential exhaustion markers, such as PD-1 and Tim-3, is associated with disease progression [9]. Patients with fatal outcomes presented increased levels of interferon-λ, TGF-α, thymic stromal lymphopoietin (TSLP), IL-16, IL-23, and IL-33, and markers linked to coagulopathy, such as thrombopoietin [81]. On the other hand, anti-SARS-CoV-2 antibodies can exert protective functions, such as neutralization, antibody-dependent phagocytosis (ADP), and antibody-dependent cellular cytotoxicity (ADCC) [29]. The Fab-mediated mechanisms include neutralization, in which the entry of the virus into the host cell is inhibited [29]. Fc mechanisms include complement activation, ADCC, and ADP. Fc-effector functions, such as ADP, have been associated with protection against coronaviruses [29]. Neutralizing antibody titers are higher in patients with severe disease than in those patients with mild disease [29]. However, antibody effector functions can also promote inflammation and amplify liver damage [29]. The pathological effects of antibodies in COVID-19 are related in part to aberrant glycosylation patterns, which are observed in the anti-SARS-CoV-2 IgG antibodies of patients with severe, but not mild, disease [82,83]. Neutrophils are the most abundant leukocytes, which express the Fc alpha receptor (FcαR/CD89) and can exhibit various effector functions, including ADP and neutrophil extracellular trap formation (NETosis) [29,84,85]. These structural modifications can trigger inflammatory processes, such as cytokine production, immune cell infiltration into some organs, or platelet-mediated thrombosis [84]. See Figure 3.

The relationship between autoimmunity and COVID-19 is complex [21,25]. Some of the clinical manifestations of COVID-19, including hyperinflammation and macrophage activation, are similar to the immunopathology of various autoimmune diseases such as juvenile idiopathic arthritis and systemic lupus erythematosus (SLE) [86,87]. Growing evidence demonstrates that the SARS-CoV-2 virus may trigger De novo autoimmunity, including SLS, and autoimmune/autoimmune-like hepatitis (AIH) [21,88,89]. Mechanistically, this could be related to virus-induced molecular mimicry, resulting in the development of new-onset autoantibodies targeting traditional autoantigens or cytokines [90]. Molecular mimicry generally occurs when microbial peptides share the same antigenic sequences with host self-proteins [89,90]. It has been demonstrated that hexapeptide sequences of SARS-CoV-2 share a similar sequence with human proteins, which can cause a wide range of complications from vascular disease to autoimmune liver disease [89,90]. Hexapeptides of N and surface glycoprotein of SARS-CoV-2 have indicated substantial sequence homology with three human proteins, namely DAB1, AIFM, and SURF1, that are involved in neuron development and mitochondrial metabolism [89,90,91,92]. Multiple trials have reported that CD4^+^ T cells and CD8^+^ T cells are active with high proportions of HLA-DR CD3, CD28, and CD38, and increase the expression of the proliferation marker Ki67 in COVID-19 [80,93]. CD8^+^ T cells have an enormous potential to eliminate SARS-CoV-2 infected cells with long-lasting immunity followed by COVID-19 [28,29]. In addition, during the clonal expansion of reactive T cells to an infection, a substantial proportion of self-reactive T cells may be increased in COVID-19 [89,90]. Cytokines can also play a critical role in the pathogenesis of autoimmune liver diseases. SARS-CoV-2 induces the secretion of proinflammatory cytokines, leading to an aberrant innate and adaptive immune response and loss of tolerance to self-antigens [89,90]. CD4^+^ T cells (Tregs) orchestrate immune homeostasis by inhibiting the proinflammatory function of effector T cells [89,90]. The number of Tregs is decreased, and they often become exhausted during COVID-19 [90]. CD8^+^ have also become exhausted upon persistent antigen stimulation, which is characterized by progressive loss of effector and proliferative potential, [28,89,90].

### 2.4. Liver Injury following COVID-19 Vaccination

The COVID-19 pandemic has resulted in enormous global morbidity and mortality rates. One of the most effective strategies for limiting COVID-19 is vaccination; this can build an immune barrier in the general population, attenuating the speed and scope of SARS-CoV-2 transmission [29,30,89,94]. Given the huge socio-economic effect of the pandemic, vaccines against SARS-CoV-2 have been developed at an unprecedented speed and scale to fight this worldwide challenge common to all human beings. In December 2020, two mRNA vaccines (BNT 162b2 Pfizer-BioNTech and mRNA-1273 Moderna, and one adenovirus (ADV) vector-based vaccine (ChAdOx1 nCOV-10 Oxford University/Astra Zeneca)) were approved by the most important drug regulatory agencies [95,96,97]. As of 4 April 2023, approximately 68% of the global population has received at least one dose anti-SARS-CoV-2 vaccine, with 13 337 398 544 doses being implemented around the world [8,89].

Several adverse events have been reported following COVID-19 vaccination, including myocarditis, vaccine-induced immune thrombotic thrombocytopenia (VITT), IgA vasculitis, and autoimmune diseases [89,94]. Autoimmune liver diseases encompass AIH, primary sclerosing cholangitis (PSC), and primary biliary cholangitis (PBC). To date, many patients with AIH following COVID-19 vaccination have been reported, however, the mechanisms involved in the development of AIH related to COVID-19 remain largely unknown. As previously mentioned, SARS-CoV-2 could stimulate autoimmunity, and vaccines could trigger autoimmune reactions [9,29]. Both the mRNA and ADV vaccines encode the intracellular production of the SARS-CoV-2 spike protein, which is the primary target for neutralizing antibodies originating from natural infection and triggering both innate and adaptive responses [89]. Through their recognition by innate intracellular sensors, including TLR 3, 7,9, and inflammasome components, vaccines induce innate immunity through cellular activation and release of FN-1, thus stimulating differentiation of CD4^+^ and CD8^+^ T cells into effector and memory subsets [89,94,98]. Although the precise mechanisms have not been elucidated, molecular mimicry is one of the major explanations of autoimmunity following vaccination [89,90,98,99,100]. Significant homology of amino acid sequences between determinants of vaccines and self-antigen may result in the synthesis of anti-spike antibodies that cross-react with structurally similar host peptide proteins [29,89,94,99,100]. Bystander activation and epitope spreading may be other mechanisms involved in the pathogenesis of AIH due to the COVID-19 vaccine [94].

The first case of AIH after the COVID-19 vaccine was reported by Brill et al. [101]. To date, most reported patients with COVID-19 vaccine-related AIH were female (83%), with a mean age of 58.7 years (range: 27–82) [89,98,102]. In 25 patients, race and ethnicity have not been reported; however, in the remaining cases, 11 patients were Caucasian, three were Asian, and 1 was Arabic [89,102]. Seventeen patients (42.5%) had a history of either liver or autoimmune disease [89,102]. One patient was three months postpartum. Most patients (78%) exhibited symptomatic AIH, with a latency time after receiving the COVID-19 vaccine of about two weeks (16.6 + 12.8 days, ranging from 2 to 60) [89,102]. Jaundice was the most common symptom (60%). Other frequent symptoms include fatigue (33%), choluria (25%), and pruritus (20%). Two patients experienced worsening symptoms after receiving the second dose. In terms of laboratory data, most patients exhibited a hepatocellular pattern of liver injury, with a substantial elevation of aminotransferases [89,103,104]. Mean GGT, ALP, and bilirubin were detected as mildly elevated. Immunoglobulin G was > 20 g/L in 13 patients (46.5%) with a mean value of 21.7 g/L. Thirty-seven patients (93%) had at least one positive antibody, and ANA was positive in 33 (83%) [89,103,104]. A liver biopsy was performed in almost all patients (98%), and the histopathological findings were typical for AIH in 3 patients and compatible with AIH in 31 patients [89,103,104]. Although five patients were receiving potentially hepatotoxic drugs, including substitutive hormonal therapy, statin, azathioprine, Peg-IFN, and nitrofurantoin, each patient had been receiving drugs for a long time without recent regimen changes [89]. The trigger vaccine was Pfizer-BioNThec in 17 patients ((43%), Moderna in 11 (28%), and Oxford AstraZeneca in 10 (20%), while CoronaVac was the trigger of AIH in two cases (5%) [89]. Steroids were implemented as a first-line treatment in 38 (95%), and azathioprine was used as a second drug in 8 patients. A decrease in laboratory data and resolution of the disease were observed in 37 of 40 patients; however, 3 patients died [89].

### 2.5. Drug-Induced Liver Injury

Drug-induced liver injury (DILI) is a further potential mechanism for aminotransferase elevation in COVID-19, either as a consequence of drugs used commonly in critical care or anti-viral drugs. DILI is a common feature, particularly in hospitalized patients with COVID-19. A recent systematic review article and meta-analysis reported a total incidence of liver injury in patients with COVID-19 to be 25.4% (95% Cl, 14.2–41.4) [105]. Liver injury has been observed in 15.2% of patients treated with remdesivir and in 37.2% of patients treated with lopinavir/ritonavir [97]. DILI was not life-threatening in COVID-19 patients [105]. Additionally, immunosuppressive drugs, such as tocilizumab, tofacitinib, and dexamethasone, used in patients with severe COVID-19 may cause liver injury through HBV reactivation in patients with occult infection [105,106]. However, randomized controlled trials investigating the efficacy and safety of remdesivir and tocilizumab in patients with COVID-19 did not indicate a significant difference in the prevalence of liver injury between the placebo group and the treatment group [107,108]. Furthermore, antibiotics and nonsteroidal anti-inflammatory drugs can also lead to liver injury in patients with COVID-19 when implemented with different indications, such as bacterial superinfection, myalgias, or fever [25].

## 3. SARS-CoV-2 Infection in Special Populations with CLD

### 3.1. Cirrhosis

COVID-19 may increase the risk of hepatic decompensation by directly causing many complications in patients with CLD. Furthermore, it also affects indirect outcomes in the management of patients with CLD, particularly in those having decompensated cirrhosis and HCC [109]. Patients with cirrhosis have an increased risk of liver decompensation and acute-on-chronic liver failure (ACLF) following viral infection, such as influenza [110]. COVID-19 in patients with cirrhosis were revealed to be associated with worsening MELD score, ACLF, and death [111]. A European registry study including 745 COVID-19 patients detected that cirrhotic patients had a significantly higher mortality rate than patients without cirrhosis (32% vs. 8%, *p* < 0.01) [112]. Similar stepwise trends have been found in the rates of ICU admission, renal replacement therapy, and mechanical ventilation. Decompensated cirrhosis was also demonstrated as an independent risk factor for death based on outcome data from patients with CLD. Another multicenter study involving 2780 patients with SARS-CoV-2 infection revealed that patients with CLD had a relatively higher mortality rate than control (relative risk (RR) 2.8, 95% confidence interval (CI) 1.9–4.0, *p* < 0.001), which was further increased in cirrhotic patients (RR 4.6, 95% CI 2.6–8.3; *p* < 0.001) [113]. A study comparing liver-related outcomes between 185 patients with CLD but without cirrhosis and 43 cirrhotic patients found a higher rate of severe liver injury and death with more advanced stages of liver disease following COVID-19 [114]. Twenty percent of patients with cirrhosis developed either ACLF or acute decompensation. A further retrospective study showed that SARS-CoV-2 infection was associated with a 2-fold increased risk of mechanical ventilation, hospitalization, and mortality in cirrhotic patients compared to noncirrhotic individuals [115]. Comorbidity and smoking increase mortality in patients with COVID-19 [116].

There are several clinical features of COVID-19 in patients with cirrhosis. Novel or worsening acute hepatic decompensation is a common presenting feature in up to 46% of patients [114]. In 20–58% of patients, this decompensation develops in the absence of typical respiratory symptoms of COVID-19 [114,115]. ACLF following SARS-CoV-2 infection is also well known, being reported in up to 12–50% of decompensating patients [18,112,114]. Registry data from Asia supports a presentation with acute decompensation or ACLF in 20% of cirrhotic patients [114]. Data from Europe also support a presentation with acute decompensation in 46% of cirrhotic patients with COVID-19, with about half of these progressing to ACLF [112]. Hepatocyte cell death is a key property of the progression to acute decompensation or ACLF in cirrhosis [26]. From a mechanistic perspective, the aberrant release of inflammatory cytokines and the activation of the inflammasome pathway in target cells may drive cell death following COVID-19 [116,117,118]. In cirrhosis, the liver further becomes vulnerable to injury, due to prior upregulation and activation of canonical and noncanonical inflammasome pathways [119]. Specifically, the SARS-CoV-2 virus has been indicated to crosstalk and activate the canonical NLRP3 inflammasome and non-canonical pyroptosis inflammasome pathways, which are both drivers of cytokine storms [118,119]. Presentation with gastrointestinal symptoms is more common in cirrhotic patients than compared to controls, and it is associated with a worse disease trajectory [120]. This phenomenon is thought to be associated with greater gut permeability and systemic inflammation [121]. Historic trials have revealed a >30-fold increase in ACE2 expression in cirrhotic vs. healthy livers, indicating that cirrhotic patients can be uniquely susceptible to COVID-19-related hepatic dysfunction [26]. Taking all data into account, patients with CLD are not at high risk of COVID-19, but these patients are at high risk of mortality [109].

### 3.2. Non-Alcoholic Fatty Liver Disease

The robust link between NAFLD and metabolic syndrome is well documented. Metabolic syndrome is also a risk factor for severe COVID-19. As such, the impact of NAFLD on COVID-19 outcomes has been closely investigated in several studies. Several observational cohorts have revealed a significant increase in the risk of severe COVID-19 in patients with NAFLD [122]. In a retrospective study involving 202 patients with COVID-19, researchers indicated that NAFLD is associated with a higher risk of progression to severe COVID-19 (OR6.4, 95% CI, 1.5–31.2), higher likelihood of abnormal liver function from admission to discharge (70% (53/76) vs. 11.1% (14/126), *p* < 0.001), and longer viral shedding time (17.5–5.2 days vs. 12.1–4.4 days, *p* < 0.001) compared to patients without NAFLD [26,122]. A multicenter retrospective study from the US reported a higher risk of ICU admission and need for mechanical ventilation but not overall mortality among patients with NAFLD-related CLD [25,123]. In addition, among younger patients (<60 years), the presence of NAFLD has been demonstrated to be associated with a >2-fold higher prevalence of severe disease [26,123]. These observations may be related to gene expression in NAFLD. Molecular studies have shown increased expression of key viral entry receptors, such as ACE2, FURIN, and TMPRSS2, in patients with NAFLD and NASH. Furthermore, ACE2 is upregulated in the liver and subcutaneous and visceral adipose tissue in obese patients with NAFLD compared to obese controls without NAFLD [21,54,124]. Collectively, this demonstrates that NAFLD in the setting of the wider metabolic syndrome likely contributes to more severe COVID-19 [21].

### 3.3. Chronic Viral Hepatitis

Several studies have examined the clinical impact of co-existing chronic HBV or HCV infection with COVID-19. A large retrospective cohort study from Hong Kong revealed that COVID-19 outcomes were no difference between patients with HBV infection and the control group including individuals without HBV infection [125]. In addition, the rates and patterns of acute liver biochemistry abnormalities during COVID-19 were the same across groups [125]. A retrospective review in Korea also indicated that chronic hepatitis B patients did not have a significantly greater risk of severe COVID-19 [126]. Additionally, those with COVID-19 had a lower rate of chronic hepatitis B than the general population, suggesting that chronic hepatitis b patients may be less susceptible to COVID-19 [126]. This protective effect may be mediated through the implementation of antiviral treatments, including tenofovir and entecavir [126,127]. Analysis from a large American Veterans dataset revealed that the rate of hospitalization was higher in HCV-positive patients with COVID-19 than those with HCV-negative [128]. However, rates of ICU admission and mortality did not differ between those with and without HCV infection [128]. Two subsequent trials have indicated adverse outcomes in patients with co-existing HCV and SARS-CoV-2, including increased ICU admissions and mortality, especially in those with high viremia [129,130].

### 3.4. Liver Transplantation

Liver transplant patients may have an increased risk for COVID-19 and a more severe clinical course because of the need for immunosuppressive drugs [25,26]. In a prospective study from Spain including a consecutive cohort of 111 LT recipients with COVID-19, researchers found that LT patients have an increased risk of COVID-19 infection but lower mortality compared with a matched general population [131]. Among immunosuppressive treatments, only mycophenolate treatment was an independent risk factor for severe COVID-19 [131]. An Italian study revealed that long-term LT patients were more prone to severe disease than short-term LT recipients, suggesting that immunosuppression per se does not increase the risk of severe disease [132]. However, the presence of chronic comorbidities typically observed in long-term recipients, such as cardiovascular diseases, kidney diseases, and diabetes, are associated with higher mortality and a more severe course of COVID-19 [132]. As in the general population, the degree of liver injury remains an independent predictor of mortality [132].

### 3.5. Hepatocellular Carcinoma

Several studies have investigated the impact of HCC in patients with COVID-19. A study including 745 COVID-19 patients indicated that the presence of HCC was not independently associated with mortality compared to patients without HCC [112]. A recent study by Kim et al. indicated, in a different cohort, that HCC is an independent predictor of death in COVID-19 patients (HR 3.31, 95% Cl 1.53–7.16). The difference between the two studies may be related to cohort size and the number of HCC patients [133].

## 4. Conclusions

The presence of liver injury is usually associated with more severe liver disease and higher mortality in patients with COVID-19. An elevated ALT level is the most robust predictor of poor clinical outcomes. Although SARS-CoV-2 proteins and viral RNA were found in tissue samples from COVID-19 patients and viral entry molecules are produced on liver cells, it is unclear whether the virus can actively infect liver cells. Liver injury and mortality in COVID-19 are likely multifactorial and result from dynamics such as a sustained and excessive release of proinflammatory and prothrombotic cytokines following the infection, iatrogenic injury, hemodynamic changes associated with mechanical ventilation or vasopressor use, and worsening the underlying injury in patients with CLD. The risk of de novo liver injury in patients without CLD is extremely limited, and, very rarely, patients who develop COVID-19-associated ACLF have been reported. Patients with NAFLD-related CLD exhibit an increased risk of severe disease, ICU admission, and need for mechanical ventilation independently of other comorbidities, including hypertension, obesity, diabetes, and cardiovascular disease. Patients with cirrhosis and COVID-19 experienced high mortality (up to 60%). Importantly, patients with liver transplantations have lower mortality when compared with the general population, although mycophenolate treatment was indicated to be an independent risk factor for SARS-CoV-2 infection.

## Figures and Tables

**Figure 1 viruses-15-01287-f001:**
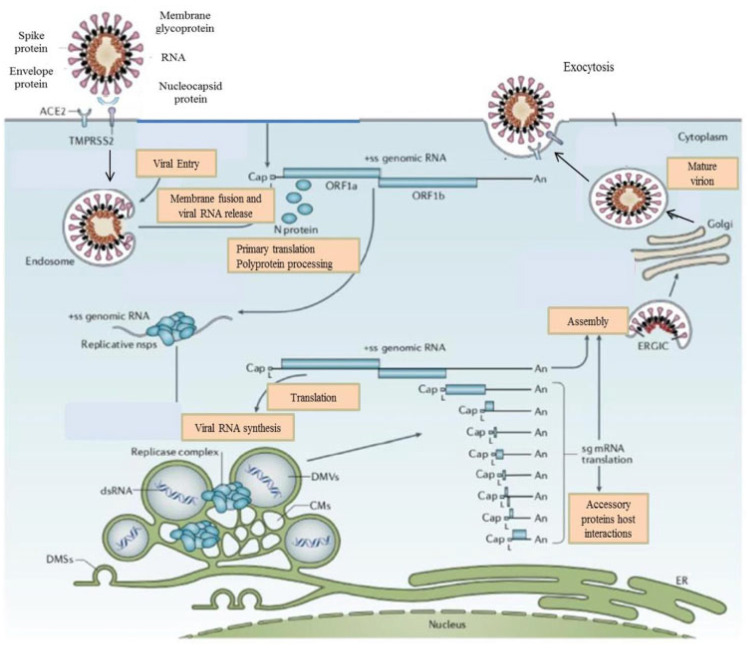
The life cycle of the SARS-CoV-2 virus. SARS-CoV-2 proteins bind to the ACE2 receptor. TMPRSS2 promotes viral uptake and fusion at the cellular and endosomal membranes. Following entry, the release and uncoating of the genomic RNA subject it to the immediate translation of the two large opening reading frames, ORF1a, and ORF1b. During the cellular life cycle, SARS-CoV-2 viruses express and replicate their genomic RNA to produce full-length copies that are incorporated into newly produced viral particles.

**Figure 2 viruses-15-01287-f002:**
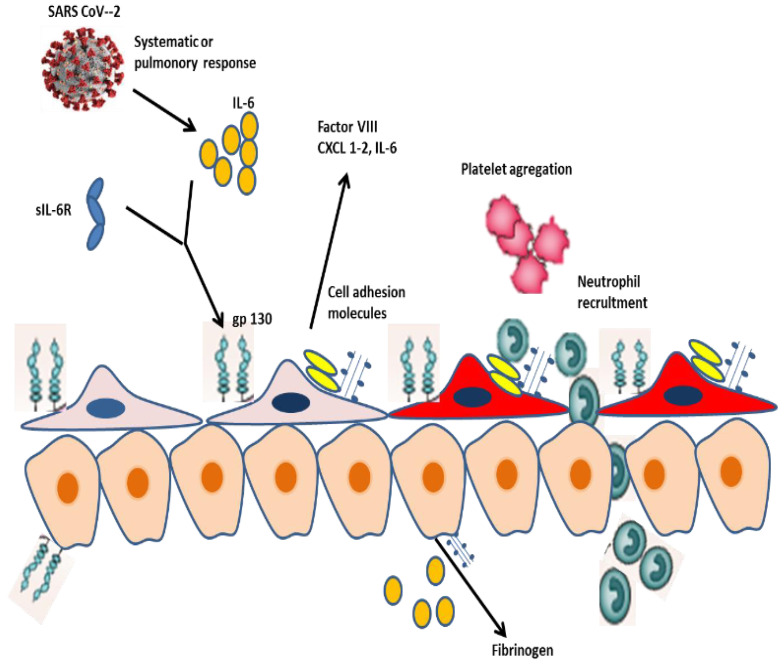
LSEC activation and cellular interaction during SARS-CoV-2 infection. SARS-CoV-2 infection stimulates a systemic release of IL-6 that induces LSECs through a trans-signaling pathway involving the sIL-6R and gp130. Activated LSECs acquired a procoagulant and proinflammatory phenotype, which express vWF, Factor VIII, CXCL1, and 2 and cell adhesion molecules, which eventually promote platelet and neutrophil recruitment in the liver. Additionally, LSECs express IL-6 and interact with hepatocytes, which respond to IL-6 through a classical signaling pathway involving the IL-6R. Hepatocytes play a critical role in the systemic response to SARS-CoV-2 via expressing fibrinogen and acute phase proteins.

**Figure 3 viruses-15-01287-f003:**
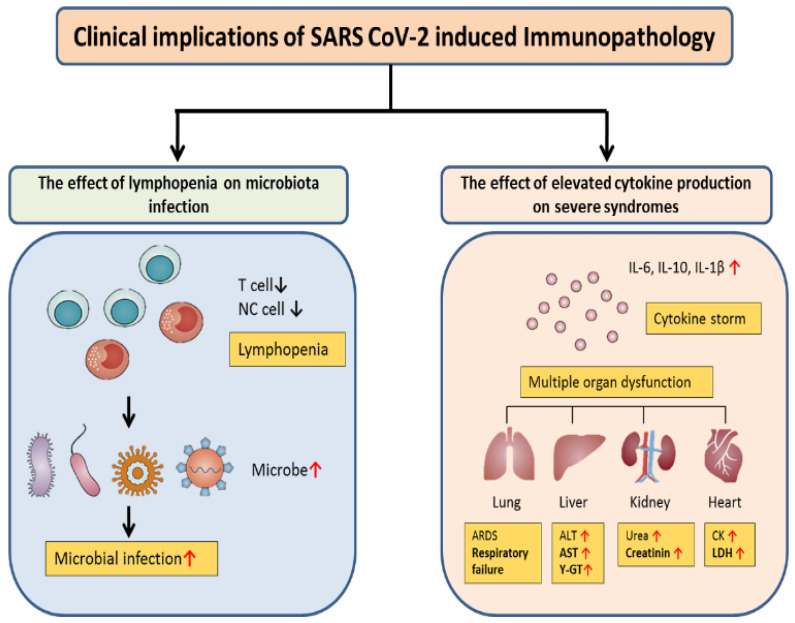
Clinical outcomes of SARS-CoV-2-related immunopathology. COVID-19 patients with lymphopenia are susceptible to infections with the microbe, which promotes disease progression and increased severity. Furthermore, cytokine storms can trigger inflammatory-associated multiple organ dysfunction, including lung injury that can result in ARDS, liver injury with ALT, AST, GGT elevation, kidney injury with increased BUN and creatinine levels, and heart injury with increased creatinine kinase (CK) and lactate dehydrogenase (LDH) levels.

## Data Availability

Not applicable.

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
