# Peer review of "Unraveling the Molecular and Cellular Pathogenesis of COVID-19-Associated Liver Injury"

_viruses, 2023, doi:10.3390/v15061287_

Round 1

Reviewer 1 Report

Dear Authors

I would like to thank you for the opportunity of reviewing this interesting paper that is focused on a very remarkable and challenging topic that is a lively argument also in the daily clinical practice.  Despite viral pneumonia representing the most common serious manifestation of COVID-19, extrapulmonary manifestations of COVID-19 have progressively gained attention due to their links to clinical outcomes and their potential long-term sequelae, especially in critically ill patients. In particular, the liver is a secondary and often collateral target of COVID-19 disease but can lead to important consequences. Therefore, papers that explore in depth this theme could surely be of interest for this important journal. Moreover, this paper demonstrates the aim of finding objective and practical conclusions from the many studies that have been conducted in recent years. This paper is pleasurable to read, although it suffers some minor limitations that Authors can easily adjust in order to slightly improve their review and making it more eligible for this important Journal. 

First of all, although language used is appropriate, I (I am not a native English speaker) recommend to Authors to obtain a certified native speaker with proficiencies in the scientific-medical field to complete properly this paper (if not yet done). 

In addition, I think references should be reformatted as suggested by Viruses Author’s guidelines in both the main text (using []) and in the references list (Author 1, A.B.; Author 2, C.D. Title of the article. Abbreviated Journal Name YearVolumepage range).

The introduction should be re-written in more scientific and organized manner, introducing also the topic of liver damage following COVID-19 infection, which is still debated. In fact, despite viral pneumonia representing the most common serious manifestation of COVID-19, extrapulmonary manifestations of COVID-19 have progressively gained attention due to their links to clinical outcomes and their potential long-term sequelae, especially in critically ill patients. Vascular complications, myocardial dysfunction, acute kidney injury, gastrointestinal symptoms, neurologic complications, and dermatologic conditions are among the reported extrapulmonary complications [doi: 10.3390/diagnostics12040846][doi: 10.1038/s41591-020-0968-3]. Furthermore, recent studies have suggested that COVID-19 could also have a serious impact on the reproductive system, altering male sperm parameters and increasing the rate of female gestational disorders, such as preeclampsia [doi: 10.1111/apm.13210]. However, whether COVID-19 could also directly affect the liver has been debated, and the literature regarding hepatic involvement in COVID-19 patients is heterogenous, due to variability in the definitions of liver dysfunction and differences in the clinical presentation and disease severity [doi: 10.1007/s12072-020-10071-9]. Despite all the evidence, in fact, the pathophysiological and immunological mechanisms of liver injury in patients with COVID-19 are still poorly understood, as well as their long-term sequelae. By stressing these concepts, the importance of the present review would be emphasized. Please cite all the aforementioned references.

Lines 39-40 are not necessary and should be removed.

Please add a title before line 42, for example: “COVID-19 pathogenic mechanism” or something like that.

At the beginning of section 2, the fact that the liver is a collateral damage of COVID-19 should be better introduced, also reporting the prevalence and incidence of this condition and discussing why these data are no homogeneous among studies. According to the current data, liver dysfunction or injury, defined as liver test abnormalities, has been reported with a prevalence of approximately 25% in COVID-19 patients, ranging from approximately 3% to 60% [doi: 10.1007/s12072-020-10071-9][ doi: 10.1007/s00535-021-01760-9][doi: 10.3748/wjg.v28.i15.1526]. These conflicting data are due to the absence of a standardized definition of COVID-19-related liver injury [doi: 10.1111/liv.14470]. In fact, despite some researchers having defined liver injury in COVID-19 patients as any liver function parameter above the upper limit of normal, others have introduced different threshold values (an increase in liver enzymes higher than 2 or 3 times the normal values) and even further classified different liver injury patterns (hepatocellular type, cholangiocytes type, and mixed type). Because of the different criteria considered, researchers may have overestimated or underestimated COVID-19-related liver damage, partially explaining the mixed results so far and jeopardizing the generalizability of the conclusions and the practical clinical implications derived from these studies. Therefore, an international consensus in this regard is urgently needed. Moreover, several studies demonstrated that patients with COVID-19 who develop liver dysfunction are mostly male, elderly, and obese [doi: 10.1002/hep.31404][ doi: 10.1016/j.jhep.2020.06.006]. Furthermore, hepatic dysfunction is significantly higher in critically ill patients, reaching up to 45% of cases, and is associated with a poor outcome, underlining its importance in clinical settings [doi: 10.1016/j.jhep.2020.04.006]. In addition, COVID-19-related liver injury is usually mild and transient, and liver failure is exceedingly rare, being reported only as anecdotical case reports in the setting of a severe disease with sepsis and coagulopathy requiring the administration of multiple drugs. Therefore, whether this drastic derangement of liver function is secondary to true viral damage rather than a bystander to the multiorgan pathophysiology of critical illness or rather the result of the hepatotoxic potential of high doses of antiviral drugs requires further discussion.

The title “4. Vascular pathologies and liver injury” should be modified, for example in “vascular alterations following COVID-19 in liver”.

Similarly, the title “5. Molecular and cellular mechanisms” should be modified, for example in “the impact of immunological and inflammatory processes of COVID-19 on the liver”, and section 6 should be incorporated in section 5 and reduced, otherwise it will sound redundant. For example, please remove lines 287-301, which seem unnecessary.

Section 8 should be emphasized since the liver is the leading site for metabolism and elimination for many drugs, including many of those being administrated for COVID-19 treatments. Therefore, drug-induced liver injury (DILI) is one important cause of liver damage in COVID-19 patients. [doi: 10.1111/apt.15916]

In section 9, please state that special populations with CLD, despite the increased susceptibility, do not appear to be at a higher risk of infection compared to other individuals in the general population [doi: 10.3390/ijms24021091]. However, the risk of mortality from COVID-19 is significantly increased in these patients (with a risk ratio of 3) [doi: 10.1038/s41586-020-2521-4], as also confirmed in a large survey including more than 17 million cases [doi: 10.1053/j.gastro.2020.04.064], especially in those with cirrhosis (where the risk ratio reaches up to 4.6), even in the absence of respiratory symptoms at the time of diagnosis. Please cite all the aforementioned references.

Finally, please add some considerations regarding patient with HCC and transplanted patients.

Best regards, 

Nicolò Brandi, M.D. 

Author Response

Dear Dr. Brandi,

I would like to thank you for your valuable suggestions and contributions in revising my manuscript.

I have summarized the revisions I made in line with your suggestions below,

1- I have revised the language of the manuscript and corrected all grammatical errors, however, I can get professional support after reviewing the revised manuscript if you wish.

2-I reformatted the references as suggested by Viruses Author's guidelines in both the main text and in the references list.

3-I rewrote ''Introduction'' section in more scientific and organized manner

a) I removed the 39-40 lines as you suggested

b)I have added a new paragraph to this section that includes the extrapulmonary manifestations of the COVID-19 infection

c)I placed the new references you suggested in the appropriate places

4) In section 2, I discussed the reasons of the wide range of reported incidence of liver injury, stating that COVID-19 causes collateral damage to the liver. I also benefited greatly from the references you recommended  in this section, and placed these references in the text.

5) I changed title 4 to be ''Vascular alterations following COVID-19 in liver''.

6) I got title 6 inside section 5 and novel title of section 5 is '' The impact of immunological and inflammatory processes of COVID-19 on the liver'' as you recommended. In this section, I removed lines 287-301. I made relevant changes and shortened the text.

7) In section 8, I underlined the importance of DILI

8) In section 9, I have benefited greatly from your review article published in Int J Mol Sci and placed the references you suggested

9) Finally, I gave information studies about patients with HCC and transplant patients who have COVID-19 infection

I look forward to receiving positive response,

Kind Regards,

Hikmet Akkız

Reviewer 2 Report

In this interesting article, Hikmet Akkız addresses a review on the pathogenesis of COVID-19 infection at the molecular and cellular level, in liver pathogenesis. This is a subject that deserves a systematic review, because the virus SARS-COV2  can promote existing chronic liver diseases to liver failure and activate autoimmune liver disease.

 The article makes an extensive review of   the direct cytopathic effects of the virus, host reaction, hypoxia, drugs, vaccination or all  these risk factors cause liver injury and has not been clarified to a large extend in COVID-19.  Also, is well updated with the latest results on immune-mediated liver injury, such as the release of pro-inflammatory cytokines such as CXCL10, CCL7, 301 and IL-1RA.

The article deals with important aspects, such as Liver injury following COVID-19 vaccination or SARS-CoV-2 infection in special populations with CLD.

Author Response

Dear Reviewer,

I would like to thank you for your valuable comments,

Kind Regards,

Hikmet Akkız

Reviewer 3 Report

The review is well written. However, there is plagiarism that needs to be fixed. Use the Grammarly software to check for plagiarism so your writing differs from previously published ones.

The review is well written. However, there is plagiarism that needs to be fixed. Use the Grammarly software to check for plagiarism so your writing differs from previously published ones.

Author Response

Dear Reviewer,

I would like to thank you for your comment regarding my manuscript.

I uploaded my manuscript to Grammarly Plagiarism Checker program as you recommended and I corrected all errors in the manuscript.

Kind Regards,

Hikmet Akkız

Round 2

Reviewer 1 Report

Authors addressed raised points appropriately.